# Carbon Nanostructures as Therapeutic Cargoes: Recent Developments and Challenges

Jagtar Singh [1], Pallavi Nayak [2,3,*], Gurdeep Singh [2], Madhusmruti Khandai [4], Rashmi Ranjan Sarangi [4] and Mihir Kumar Kar [5]

[1]  Department of Pharmacology and Toxicology, National Institute of Pharmaceutical Education and Research, SAS Nagar, Mohali 160062, India
[2]  School of Pharmaceutical Sciences, Lovely Professional University, Phagwara 144411, India
[3]  St. Soldier Institute of Pharmacy, Jalandhar 144001, India
[4]  Royal College of Pharmacy and Health Sciences, Berhampur 760002, India
[5]  Sri Jayadev College of Pharmaceutical Sciences, Naharkanta, Bhubaneswar 752101, India
*  Correspondence: pallavinayak97@gmail.com; Tel.: +91-865-801-9502

**Abstract:** Recent developments in nanotechnology and process chemistry have expanded the scope of nanostructures to the biomedical field. The ability of nanostructures to precisely deliver drugs to the target site not only reduces the amount of drug needed but also reduces systemic adverse effects. Carbon nanostructures gained traction in pharmaceutical technology in the last decade due to their high stability, ease of synthesis, tunable surface chemistry, and biocompatibility. Fullerene, nanotubes, nanodiamonds, nanodots, and nanoribbons are among the major carbon nanostructures that have been extensively studied for applications in tissue engineering, biosensing, bioimaging, theranostics, drug delivery, and gene therapy. Due to the fluorescent properties of functionalized nanostructures, they have been extensively studied for use as probes in cellular imaging. Moreover, these nanostructures are promising candidates for delivering drugs to the brain, bones, and deep-seated tumors. Still, research gaps need to be addressed regarding the toxicity of these materials in animals as well as humans. This review highlights the physicochemical properties of carbon nanostructures and their categories, methods of synthesis, various techniques for surface functionalization, major biomedical applications, mechanisms involving the cellular uptake of nanostructures, pharmacokinetic considerations, recent patents involving carbon-based nanostructures in the biomedical field, major challenges, and future perspectives.

**Keywords:** carbon nanostructures; green chemistry; drug delivery; biomedical applications; theranostics





## 1. Introduction

The discovery of a 0-D carbon allotropic structure called fullerene in 1985 by Robert Curl, Harold Kroto, and Richard Smalley was a breakthrough in the field of materials science [1]. These unique carbon hollow structures have gained traction in biomedical research. The fine-tuning of carbon atoms to form nanostructures such as carbon nanotubes, carbon nanoribbons, carbon nanodots, carbon diamonds, carbon onions, graphene nanoparticles, etc., were explored by materials scientists for potential applications in targeted drug delivery, bioimaging, tissue engineering, and so on [2]. The physicochemical characteristics of carbon nanostructures, such as mechanical, optical, and electrical properties and surface-to-volume ratio, offer the potential for desired surface modification and functionalization. Thus, carbon-based nanostructures have been extensively explored for tumor targeting, biosensing, and theranostics [3].

These nanostructures comprise $sp^2/sp^3$ hybridized carbon atoms bonded to form cage-like geometries. The size and shape of these structures can be fabricated in accordance with the amount of drug to be loaded and the target site [4]. The size of carbon nanostructures can range from 1 to 100 nm. Hybridization of the structures with other polymers has also been

explored to create nanocarriers with distinctive properties, as they are more biocompatible and selective, and less toxic [5]. Carbon nanostructures with surface-functionalized moieties can be loaded with drug molecules. These carrier systems, when targeted to the site of interest, are taken up by receptor-mediated endocytosis or direct penetration into deep-seated tumors due to their high tissue permeability [6,7]. Thus, carbon nanostructures provide promising insights into developing novel formulations for selectively targeting the CNS, hepatocytes, bone marrow, cancer cells, and so on [8]. Unlike liposomal preparations, the ease of synthesis, functionalization, and higher stability of carbon nanostructures make them materials of interest for developing targeted drug therapies [9].

This review explores the fundamental properties, synthetic pathways, functionalization, and various architectures of carbon nanostructures. A comprehensive description of the biomedical applications for each of these nanostructures is also presented, along with a review of current challenges and future perspectives.

## 2. Properties of Carbon Nanostructures

Different carbon-based nanomaterials make up the carbon family. Zero-dimension structures, such as fullerene and carbon dots, one-dimensional carbon nanotubes (CNT), two-dimensional graphene, and nanodiamonds (NDs), are examples of carbon-based nanomaterials (NDs) [10]. Carbon nanocages can be created by reducing the size of one-dimensional carbon structures down to the nanoscale level [11]. Similar to macroscopic allotropes, the characteristics of a particular carbon nanostructure depend on the type of hybridization that the nanostructure's carbon atoms adopt [12]. Since carbon nanostructures are made entirely of carbon, they have high stability, excellent conductivity, low toxicity, and are environmentally friendly. Their mechanical, thermal, and electrical conductivities are distinctive [13]. The Young's moduli and tensile strength for carbon nanotubes and graphene (GR) are in the ranges of 1 TPa and 130 GPa, respectively. The thermal conductivities of CNTs and GR are around 3000–3500 W/mK and 5000 W/mK, respectively [14]. CNTs and GR have promising potential to improve electrodes for neural interfaces because they are naturally excellent electrical conductors with adjustable biocompatibility. According to one theory, carbon-based nanomaterials could be used to create artificial scaffolding that would interface with neuronal activity and encourage neuroregeneration, for instance, after spinal cord injuries or other brain disorders [15]. This could only be obtained with particular types of carbon nanostructures, as such scaffolding requires unusual physical properties, exceptionally high mechanical strength, and electrical conductivity [4]. Since carbon nanostructures are hydrophobic in nature, formulation development poses a great challenge. Surface modification, functionalization, co-polymerization, and the use of surfactants have all been investigated as ways to improve the dispersion of carbon nanostructures in water and other appropriate solvents [16]. In addition, the tuning of various carbon nanostructures with smart bio-responsive polymers, such as magnet-sensitive, redox-sensitive, light-sensitive, pH-sensitive, and electrolyte-sensitive polymers, can achieve better drug targeting in in vitro and in vivo conditions [17]. Figure 1 summarizes the many features of carbon nanostructures that enable their biomedical applications.

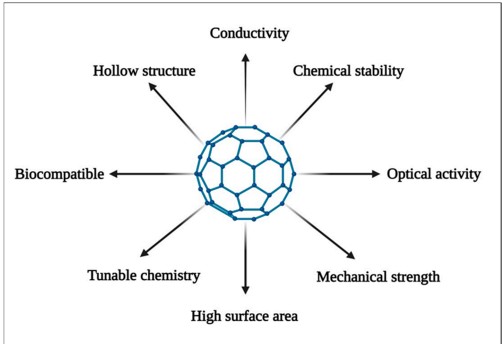

**Figure 1.** Properties of interest for employing carbon nanostructures in biomedical research.

### 3. Green Approaches for Synthesis of Carbon Nanostructures

The synthesis of nanomaterials involves the generation of chemicals that are toxic to animals, humans, the environment, and microbiota. Moreover, the raw materials required for the synthesis of these nanostructures are toxic themselves [18]. Therefore, it is necessary to develop new green routes for synthesis of nanomaterials. Chemistry is the emerging field of science focusing on reducing the use and generation of hazardous chemicals by developing sustainable and eco-friendly technologies. These technologies are not only safe and cheap but also find a useful application in nanomaterial synthesis, with high yield and economic value [19].

Plants have extensively been explored for their application in the green synthesis of carbon nanomaterials. Plants contain a variety of nutritious compounds that come from the different parts of the plant, including the leaves, fruits, stems, bark, roots, and seeds such as steroids, flavonoids, alkaloids, flavones, polyphenols, saponins, tannins, terpenoids, and other compounds [18]. Secondary metabolites present in plant extracts act as lowering, stabilising, and capping agents for bioreduction reactions involved in nanomaterial synthesis. The antioxidants and hydrophilic components included in these plant extracts address the issue of carbon nanoparticle long-term stability and solubility [20]. In 2017, Tripathi et al. [21] prepared carbon nanotubes from extracts of *Juglans regia* (walnut), Rosa (rose), and *Azadirachta indica* (neem) at a temperature of 575 °C using thermal chemical vapor deposition [21]. The CNTs thus formed had diameters in the range of 8–15 nm. In 2020, Damera et al. [22] prepared fluorescent carbon nanoparticles by hydrothermal technology from eucalyptus twigs for bioimaging [22].

Apart from plants, waste-derived production of carbon nanomaterials has also been explored by chemical engineers. Bio-sludge, e-waste, plastic, rubber tires, and shavings from muffle furnaces are among the significant but least explored sources for synthesizing carbon nanomaterials [23]. In 2017, Maroufi et al. [24] reported the synthesis of carbon nanoparticles from waste rubber tires under high-temperature conditions (>1550 °C). The decomposition of waste rubber tubes led to the formation of carbon nanoparticles with a specific surface area of 117.7 $m^2/g$ and a diameter of 30–40 nm. In another study, carbon nanotubes were synthesized from plastic waste (polyethylene tetraphthalate). The rotating cathode arc discharge technique was employed to prepare multi-walled carbon nanotubes at a temperature of approximately 2600 °C [25].

In 2018, Liu et al. [26] developed fenbufen-loaded single-walled carbon nanotubes. As binary green solvents, 1-butyl-3-methylimidazoliumtetrafluoroborate and choline chloride/ethylene glycol were utilized. The functional monomer was 4-vinylpyridine (4-VP), while the cross-linking monomer was ethylene glycol dimethacrylate. The action of this drug-loaded nanostructured polymer was regulation of fenbufen release and improved overall bioavailability. Table 1 lists some recently developed green approaches for the fabrication of carbon nanostructures.

**Table 1.** Green routes for the synthesis of carbon nanostructures.

| Nanostructure | Source of Carbon | Green Pathway | Application | Reference |
|---|---|---|---|---|
| Quantum dots | Green tea | Hydrothermal | Photodynamic therapy | [27] |
| Quantum dots | *Zingiberis rhizoma* | Pyrolysis | Analgesic | [28] |
| Nano onions | Carbonization | Tomatoes | Theranostics | [29] |
| Nanodiamonds | Coal | Laser ablation | Bioimaging | [30] |
| Nano horns | UV/$H_2O_2$ oxidation | Cellulose | Drug delivery | [31] |

## 4. Classification of Carbon Nanostructures

Carbon-based nanostructures are classified on the basis of dimensional characteristics (0-D, 1-D, 2-D, and 3-D) and allotropic forms (fullerene derivatives, graphene derivatives, and other amorphous forms), as presented in Figure 2.

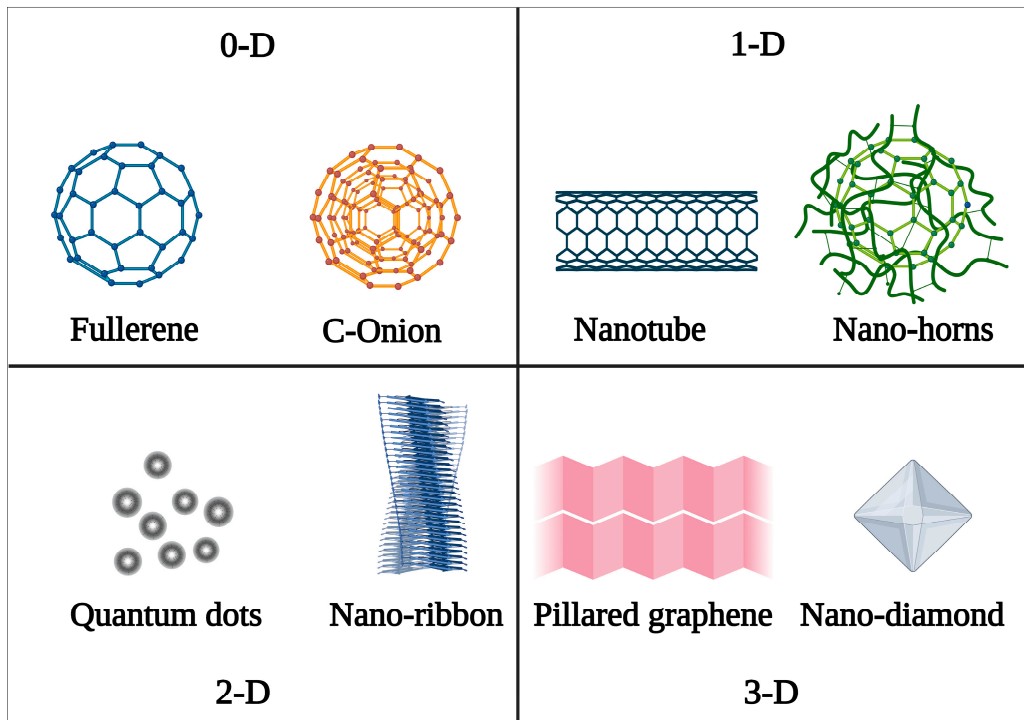

**Figure 2.** Various architectural categories of carbon nanostructures.

### *4.1. Fullerene*

The structure elucidation of fullerenes reveals that these are closed, hollow cages with $sp^2$ hybridized carbon atoms. However, these cages cannot be produced only from hexagonal rings. Fullerenes, buckminsterfullerene, and $C_{60}$ are examples of endohedral fullerenes, often known as buckyball clusters or buckyballs. Each of these compounds contains fewer than 300 carbon atoms [32]. Euler's theorem can be used to show that a spherical surface must have exactly 12 pentagons [33]. This group of fullerenes, which are closed-cage carbon compounds, is identified by the symbol Cn, where n stands for the total number of carbon atoms. Fullerenes have 12 pentagons and a variable number of hexagons. Depending on the number of hexagons involved, different sizes of fullerenes can be synthesized. There are two alternative values for n: $n = 20$ or $n = 20 + 2k$ (k = 1, 2, 3, etc.) [34]. Fullerenes provide a number of functional sites that make it possible to bind chemical components of targeting ligands in the three-dimensional structure, which makes it easier to target cells. In addition to this, it is possible to optimize their allotropes, pharmacokinetic properties, therapeutic results, and other characteristics such as size, hydrophilicity, and colloidal stability in an environment that mimics the body's natural conditions [35].

This molecule's low solubility in many organic solvents and insolubility in water present challenges for its use in pharmaceutical applications. Different approaches have been employed to make these structures more hydrophilic and water-dispersible [36]. The preparation of two-phase colloidal solutions, fullerene co-polymerization, derivatization of fullerenes, encapsulation in smart polymers (polyvinylpyrrolidone, cyclodextrins, micelles, calixarenes, liposomes, etc.), and chemical modification (addition reactions with carboxylic acids, amino acids amphiphilic polymers, and polyhydroxyl groups (fullerenols)) are a few strategies for enhancing the dispersibility of carbon nanostructures [37].

### 4.1.1. Synthesis

Pyrolysis of higher polycyclic molecules, such as naphthalene, corannulene, or polycyclic aromatic hydrocarbons, can be employed to synthesize fullerenes. In the presence of argon or other inert gases, these compounds break down through the cleavage of hydrogen bonds at high temperatures (approximately 1000 °C), producing $C_{60}$ and $C_{70}$ [29]. Another method for producing fullerenes uses an arc discharge. Two graphite electrodes are subjected to a high voltage, and the discharge causes the graphite to vaporize, resulting in plasma. Fullerenes are produced when graphite plasma condenses into particles which are then deposited on the reactor walls [38]. Another method involves the resistive heating of carbon rods under a partial helium atmosphere. As a result, the carbon rods release a thin, white plume that mimics soot and is gathered on glass shields that surround the carbon rods [39].

### 4.1.2. Functionalization

The use of functionalized fullerenes in therapeutics and diagnostics is constantly expanding. They primarily use functionalized derivatives for quenching reactive oxygen species, targeted drug delivery, and bio-imaging [40]. Surface functionalization is an effective method that can be utilized to render fullerene soluble and dispersible in organic as well as water-based solvents. There are two chemical processes that can be utilized to modify the fullerene surface: (A) complexation with a solubilizing agent to partially cover the lipophilic surface of fullerene, and (B) covalent surface-functionalization. Both of these processes are described in more detail below. A typical method for the functionalization of $C_{60}$ involves grafting amine groups onto the molecule by first combining it with a variety of primary aliphatic amines, such as n-propylamine, t-butylamine, and dodecylamine [41]. The Prato reaction of an azomethine ylide with a $C_{60}$ molecule produces a stable product that grafts a pyrollidine ring to $C_{60}$. This is another common amination process [42].

### 4.1.3. Applications
Biosensing

Fullerenes have been found to be stable, biocompatible, highly responsive to biochemical reactions, and have very fast response times. $C_{60}$ compounds are employed to develop biosensing techniques [43].

In 2022, Kurbanoglu et al. [44] used conjugated polymers and fullerenes to create a fullerene-based electrochemical tyrosinase (Tyr) enzyme inhibition biosensor for the drug indomethacin (INDO). Three conjugated polymers including benzoxadiazole, thienopyrroledione, and benzodithiophene moieties were employed for this, along with fullerene as a transducer modifier for catechol detection. A highly sensitive and quick-response catechol biosensor could be made using a specific mix of these compounds. The effect of the chronoamperometric measurement parameters on the biosensor response was investigated. Catechol biosensing was achieved at concentrations ranging from 0.5 to 62.5 μM, with a detection limit of 0.11 μM. Tyr inhibition was measured, and INDO was found to have an I50 value of 15.11 M and a mixed type typical of enzyme kinetics [44].

The pharmacokinetic response in specific individuals varies, which results in non-responsiveness for a specific reason. One of the major reasons could be the presence/alterations/deficiencies of the genes encoding the enzymes that metabolize these drugs. The detection of such genetic alterations can be achieved by fullerene-based biosensors. People with the CYP2C19*2 gene require more doses of clopidogrel [12]. Therefore, to detect the presence of this gene in humans, Zhang and colleagues 2018 [13] constructed an ultrasensitive electrochemical biosensor (c$C_{60}$/$CeO_2$/PtNPs). To generate the signal probe, the fullerene-based nanoparticles were labeled with the signal probe. Following the sandwich reaction of the CYP2C19*2 gene between the capture probe and the signal label, a distinct electrochemical signal resulting from the signal label's catalysis of hydrogen peroxide would be detected. Amperometry was used to record electrochemical signals. The approach demonstrated a strong linear relationship between the current and the logarithm

of CYP2C19*2 gene concentrations in the 1 fM to 50 nM range, with a limit of detection (LOD) of 0.33 fM (S/N = 3). When compared to possible interference-causing chemicals, the suggested approach demonstrated high specificity to target DNA [13,44].

Drug Delivery

Fullerene is a carbon compound of interest due to its biocompatibility. Its challenging aspects, nevertheless, are the safety concerns and the desire to increase drug loading. These issues have been addressed by using a variety of techniques, such as super-positioning the hydrogen bonds, stabilization with an OH group, and functionalization with an OH group [39,45].

In 2022, Giannopoulos et al. [46] developed a molnupiravir-loaded fullerene-based nanocarrier system, since molnupiravir has been shown to be quite effective in reducing the duration of hospitalization related to COVID-19. This formulation had two molnupiravir molecules externally linked to carboxy fullerenes, known as dendrofullerene. Two correctly produced nitrogen single bonds (N-N) were used as linkers between the dendrofullerene and the two molnupiravir molecules to create the final structure of the $C_{60}$ derivate/molnupiravir conjugate. This resulted in the synthesis of a drug system with increased water-solubility that released the drug in a synchronized manner to the target organs [46].

Raza et al. (2015) [47] studied the possibility of $C_{60}$ fullerene use for docetaxel delivery to malignant cells. The drug was carboxylated, acylated, and conjugated to $C_{60}$ fullerenes. The nanoconstruct developed boosted docetaxel bioavailability by 4.2 times and lowered drug clearance by 50%. This system regulated the release of drugs and was shown to be beneficial to erythrocytes. The cytotoxic capability of the tested MCF-7 and MDA-MB231 cell lines were likewise amplified many times, demonstrating greater efficacy with lower doses.

In photodynamic cancer therapy, certain photosensitizers are used for generating reactive oxygen species in the tumor microenvironment [47]. These ROS elicit a potent cytotoxic effect on cancer cells while preserving the physiology of adjacent normal cells. The target specificity and hydrophobicity of the photosensitizers provide a significant challenge in cancer photodynamic therapy [48]. Many techniques have been explored to conjugate carbon nanostructures with photosensitizers. Iron oxide nanoparticles (IONPs) were first deposited onto the surface of fullerenes before being PEGylated. A potent PDT capacity and strong superparamagnetism were shown by the $C_{60}$-IONP-PEG nanocomposite [30,49].

$C_{60}$-lysozyme demonstrated effective endogenous reactive oxygen species generation in a different investigation. Due to its photodynamic activity in HeLa cells, exogenous $H_2O_2$ production causes activity afterward [50,51]. Table 2 summarizes some recent advances in fullerene-based drug delivery systems.

**Table 2.** Recent developments in fullerene-based drug delivery systems.

| Nano System | Functionalized With | Technique Used | Drug Used | Application | Reference |
|---|---|---|---|---|---|
| Heterofullerene $BC_{59}$, $SiC_{59}$, and $AlC_{59}$ | Pristine $C_{60}$ and heterofullerene | Conjugation | Hydroxyurea | Treatment of myeloid leukemia | [52] |
| Al and Si doped fullerene | Pristine $C_{60}$ and chloroquine | Chemical interaction | Chloroquine | Enhanced efficacy against COVID-19 | [53] |
| $C_{60}$-Dex-$NH_2$ | TPFE | Conjugation | siRNA | Treatment of prostate cancer | [31,54] |
| $C_{60}$-nano complex | Cisplatin | Conjugation | Cisplatin | Enhanced penetration in tumor cells | [55] |
| $C_{60}$-DOX | DOX-complex | Covalent interaction | Doxorubicin | Enhanced in vitro cytotoxicity in cancer cells | [56] |

*4.2. Carbon Nano-Onions*

Carbon nano-onions (CNOs) are zero-dimensional carbon nanoparticles that may be identified by their tightly closed, onion-like shells that have multiple layers and enclose one another entirely. These CNOs have a $C_{60}$ or $C_{80}$ fullerene at their center, an interlayer distance of about 3.4 Å, and a diameter of between 1.4 and 50 nm [8]. The carbon atoms in the carbon onion have undergone $sp^2$ hybridization. Due to the intrinsic $sp^2$ surface of CNOs, covalent and non-covalent functionalization is simple, and various functional groups can be attached [57]. Recently developed CNOs have a wide range of uses, including their use as photothermal agents in photothermal cancer therapy and use of their cargo-like characteristics for drug administration. Similar to fullerenes and other carbon family nanostructures, carbon nano-onions also lack water dispersibility, but can be made water dispersible upon surface functionalization and employed for drug delivery [58].

4.2.1. Synthesis

Carbon nano-onions are synthesized by various methods to obtain structures with the desired surface area, size, shape, oxygen content, density, shell count, and conductivity. Electron beam irradiation, ball milling of graphite, chemical vapor deposition, underwater arc discharge, ion-implantation, various plasma procedures and pyrolysis, and thermal annealing are among some commonly used techniques for the synthesis of carbon nano-onions [59]. Carbon nano-onions of 30–50 nm are produced via the pyrolysis of carbon nano-diamonds at 1000–1500 °C in a vacuum [28]. A novel method for the synthesis of CNOs was developed by A. Guo et al. in 2021, who reacted hydrothermal citric acid at 180 degrees Celsius in the presence of $KNO_3$. According to the TEM-based formation process, there were three stages of reaction: the dehydration of citric acid into graphene quantum dots (GQD); the growth and piling into graphitic nanosheets (GS), which roll up into graphitic hollow polyhedrons (GHP) due to interface energy; and then the disordering and rearrangement into spherical CNOs with the lowest interface energy in the hydrothermal solution. All of these stages took place sequentially [60]. To prepare CNOs, thermal annealing of DNDs is currently thought to be the best approach. This method has the largest potential for industrial mass production since it produces CNOs that are narrowly distributed and extremely pure [61].

4.2.2. Functionalization

Different methods, such as radical addition, cyclopropanation, 1,3-dipolar cycloaddition, fluorination, reduction, alkylation, and oxidation, can covalently functionalize the $sp^2$ surface of CNOs. Targeting, bioimaging, and biosensing compounds are often covalently bonded to nanocarrier surfaces [62]. Another example is the click reaction between an azide- and acetylene-functionalized CNO molecule [50,63]. Non-covalent interactions of importance to CNO-based nanocarriers include stacking, charge, lone pair, polar, and dipole interactions, as well as hydrophobic and dipole-induced dipole interactions. Van der Waal forces also extend to charge-dipole interactions, dipole-dipole interactions, and dipole-induced dipole interactions [64]. The covalent functionalization, specifically by oxidation, increases their water dispersibility and expands the potential for amidation and esterification processes [28,65].

4.2.3. Applications
Biosensing

In 2022, Sharma et al. [66] developed an ex situ mixing technique to create a ZnO/CNO nanocomposite. The electrochemical sensing response of the ZnO/CNO nanocomposite for a linear range of glucose concentrations (0.1–15 mM) was investigated using cyclic voltammetry. The ZnO/CNO nanocomposites showed greater glucose sensing capabilities with a sensitivity value of 606.64 A/mM cm$^2$. Using this technology, non-invasive skin-attached sensors for biomedical purposes could be created [66].

Carbon nano-onions (CNOs) were used as supports by Sok and Fragoso (2018) to immobilize alkaline phosphatase, horseradish peroxidase, and glucose oxidase [67]. Initially, carboxylic groups were produced on the surface of CNOs by oxidation, which was followed by covalent coupling using a soluble carbodiimide. According to thermogravimetric analysis, the specific enzyme load contained roughly 0.5 mg of protein for every mg of CNO. Without changing to the ideal pH and temperature, the mounted enzymes demonstrated improved storage stability. This developed nanobiocatalyst is potentially interesting for biosensing and other biotechnological applications because of these characteristics [67].

Drug Delivery

Carbon nano-onions, when functionalized, can be made water dispersible and utilized for site-specific delivery of drugs. Their low toxicity and minimal inflammatory potential make the CNOs suitable for targeted drug distribution. They are a promising drug carrier due to their enormous surface area and readily available pi-electrons. For pH-responsive drug release, in 2019 Mamidi et al. [58] developed zein protein hydrogels infused with poly 4-mercaptophenyl methacrylated CNOs (f-CNOs). In order to create the nanosystem, oxidized CNOs were functionalized using 4-mercaptophenol, then methacrylated and polymerized to produce f-CNOs. Acoustic cavitation was used to integrate this nanomaterial into a zein protein matrix. Investigations into its drug release indicated that the system exhibited a pH-sensitive release pattern, with the quickest release occurring between pH 7.4 and 9.0, making it a promising candidate for drug therapy against colorectal cancer.

In 2021, Ahlawat et al. [68] developed acetylcholinesterase inhibitor-loaded CNOs to increase the efficacy of an anti-Alzheimer's drug in crossing the BBB and to release the drug in a sustained manner at a pH of 7.4. The size distribution of these nanostructures ranged from 1 nm (35%) to 15 nm (65%). Animal models were used to assess the ability of these water-soluble CNOs to penetrate the BBB and inhibit the acetylcholinesterase enzyme. These CNOs were found to be rapidly eliminated within six days of administration rather than accumulating in the brain, as shown by the gradually rising concentration of CNOs in excreta [68]. Some recently investigated carbon nano-onion systems for drug delivery are given in Table 3.

**Table 3.** Recent developments in CNO-based drug delivery systems.

| Nanosystem | Drug Used | Application | Reference |
|---|---|---|---|
| Polycaprolactone/fts i nanocomposite fibers | Doxorubicin | pH dependant release of drugs at the tumor site | [69] |
| f8DIN CSL_CIT composite hydrogels | 5-FU | Sustained and site-specific release of 5-FU | [70] |
| HADIN CSL_CI | HA-Phospholipid | Targeting of overexpressed CD44 cancer cells | [71] |
| CTAB-CNOs | CTAB | Antimicrobial action against *Escherichia coli)* | [72] |
| PHPMA-CNOs = f-CNOs | Doxorubicin | Thermosensitive, prolonged, and site-specific release of drug | [73] |

### 4.3. Carbon Quantum Dots

Carbon quantum dots (CQDs) are the newest member of the nanomaterial family, possessing ease of synthesis. CQDs were serendipitously discovered while processing single-walled carbon nanotubes (SWCNT) [10,74]. Since then, CQDs' fluorescence characteristics have been thoroughly investigated as fluorescent biocompatible-nanostructures. With a particle size of less than 10 nm, CQDs are quasi-spherical carbon nanostructures. CQDs are spherical zero-dimensional allotropes that exhibit a number of intriguing characteristics, including biocompatibility, environmental friendliness, conductivity, optical

qualities, water solubility, and low toxicity [74]. When exposed to light, these chemicals generate reactive oxygen species (ROS), which can kill bacteria and hence treat illnesses [75]. Both crystalline and amorphous CQDs are possible. In CQDs, the carbon hybridization is $sp^2$, while $sp^3$ hybridization has also been documented in a few instances. CQDs' crystal lattice parameters are around 0.34 nm [76]. CQDs are administered parenterally, as a minimal dose is required, and a larger volume of distribution can be achieved throughout the body. Depending on the target tissue of interest, the surface of a CQD is functionalized with particular ligands for their preferential uptake by specific tissues [77].

### 4.3.1. Synthesis

The synthesis methods for CQDs can be broken down into two categories: top-down and bottom-up. The top-down method involves breaking down a larger bulk item or nano-material into particles smaller than 10 nm [78]. High-energy ball milling, electrochemical synthesis, and laser ablation are all examples of top-down methods for creating CQDs. While solvothermal, pyrolysis, reverse micelles, microwave-assisted synthesis, hydrothermal, combustion, ultrasonic, chemical vapor deposition (CVD), and others are examples of bottom-up processes for making CQDs, which refers to carbonization or synthesis from tiny molecules [79]. Controlling the size of CQDs can be accomplished through post-synthesis changes such as filtering, dialysis, sonication, centrifugation, column chromatography, and gel electrophoresis [73].

The low-cost industrial synthesis of CQDs is achieved from pure graphite electrodes, electrochemically exfoliated in a variety of pH electrolytes, including $H_2SO_4$, NaCl, and NaOH, to produce CQDs at a low cost [80]. Another method for producing carbon gas plasma with an intense laser pulse in organic liquids is laser ablation. A Q-switched ND:YAG laser system is used to irradiate natural graphite flakes that are suspended in an ethanol and diethylenetriamine solution. N-doped quantum dots resembling graphene are created when the laser-induced plasma plume condenses, and they grow to a size of roughly 6 nm [81].

### 4.3.2. Functionalization

Surface functionalization has a considerable impact on the carbon dots' absorbance and photoluminescence characteristics [82]. CQD functionalization affects other properties by changing their ability to interact with drugs, other chemical molecules, ions, and bodily organs. As a result, surface functionalization (for example, cell internalization, cell local-ization, and cytotoxicity) is one of the most important possibilities to consider in terms of CQD biological applications [34]. CQDs can have their surfaces functionalized with a wide variety of macromolecules, any of which may be electrically neutral or charged (positively or negatively). Polyethylene glycol is an example of a macromolecule that does not have a charge (PEG) [83]. PEG works to block an immune response by inhibiting protein binding that is not specific to any one protein. Additionally, it is biodegradable and compatible with living organisms. CQDs can be positively charged by cationic macromolecules such as polyethyleneimine; therefore, these CQDs are able to bind to proteins with a nega-tive charge that are found in cell membranes (PEI). Therefore, positively charged CQDs have an effect on the structural integrity of cell membranes, which makes cell transfection possible [84].

On the surface of CQDs, there are numerous functional groups (-OH, -COOH, -NH2, etc.) that can serve as active coordination sites for transition metal ions. Electrocatalytic performance may be improved by heteroatom doping on CQDs with other inorganic substances such as metal phosphides and metal sulfides [85].

### 4.3.3. Applications
Bioimaging

CQDs are crystalline semiconductors with optical and electrical characteristics. Thus, CQDs can serve best for designing point-of-care-testing (POCT) diagnostics. The detection

of early biomarkers of neurodegeneration, cardiac ischemia, and diabetic pathologies can help in timely decision-making for precise therapeutic interventions [86].

The most advantageous and noticeable characteristic of CQDs is their photoluminescence. The chemistry and the quantum confinement of CQDs are the two key factors that influence their optical characteristics. The two leading bands in the usual absorption spectrum of CQDs are located at about 230 and 350 nm [87]. Drug delivery can be visualized using CQDs. Feng et al. developed a nanocarrier based on cisplatin(IV) prodrug-loaded and charge-convertible CQDs [CQDs-Pt(IV)] for image-guided drug delivery in 2016. CQDs-Pt(IV) were coated with an anionic polymer containing dimethylmaleic acid, which showed a charge conversion in the tumor's slightly acidic extracellular environment (pH = 6.8). Furthermore, the positive charges on the nanocarrier made it simpler for it to pair with and internalize in the negatively charged cellular membranes. CQDs-Pt(IV) nanocarriers were used in in vivo studies that revealed strong tumor-inhibition efficacy and few adverse effects [88].

Another study conducted by Gao et al. in 2017 [89] used a fluorescent CQD that had been turned on as a theranostic nanoprobe. CQDs (P-CQDs/HA-Dox) were made by combining doxorubicin-conjugated hyaluronic acid with a carbon dot modified with polyethyleneimine. The functionalized P-CQDs/HA-Dox served as an effective tool for self-targeted imaging, hyaluronidase detection, and drug delivery. By explicitly focusing on the CD44 receptors that are overexpressed in cancer cells, the resulting nanoprobes improved internalization. Once within the cells, hyaluronidase was activated, causing the HA-Dox to break apart and release doxorubicin, which resulted in the P-CQDs' fluorescence [89]. Some recently explored bioimaging techniques employing CQDs are given in Table 4.

**Table 4.** Recent developments in CQDs-based bioimaging techniques.

| Nanosystem | Method | Application | Reference |
|---|---|---|---|
| CQD-RhB-Si Nps | Surface functionalization | In vivo imaging of $Cu^{2+}$ in body tissues | [34] |
| N-P-CQDs | In situ hydrothermal synthesis | Imaging of $Fe^{3+}$ ions and cell imaging | [90] |
| FA-CQDs | Solvothermal process | Fluorescent imaging of blood vessels | [91] |
| B1,B2-CQDs | Thermal carbonization | Imaging of myoblasts (mouse satellite) | [92] |
| N-CQDs | Hydrothermal process | Fluorescent staining of *E. coli* and *S. aureus* | [93] |

Drug Delivery

Due to the unique characteristics of CQDs, including small size, water solubility, high cell membrane permeability, low toxicity, fluorescence emission, chemical inertness, ease of production, possible functionalization, and drug loading, CQDs have drawn increasing attention as a means of drug delivery.

Nitrogen-containing CQDs were covalently bonded to 7-(3-bromopropoxy)-2-quinolylmethyl chlorambucil (Qucbl) by Karthik et al. (2013) [94]. An in vitro study revealed that the cytoplasm and nucleus of the drug-loaded CQDs were colonized. The drug-loaded CQDs controlled drug release in order to irradiate and destroy cancer cells [94]. Photoluminescent features can also be used as an adjuvant to cancer cell drug therapy. In 2015, Beack et al. [95] reported that a CQD-chlorine e6-hyaluronate (CQD-Ce6-HA) molecule was also efficient in the photodynamic therapy treatment of mouse melanoma. Thermal decomposition of glycerin resulted in CQDs, which were then conjugated with Ce6, a benign photosensitizer with a strong singlet oxygen generation. For targeted delivery to cancer cells, hyaluronic acid was added to the CQD-Ce6 combination. As a result of inducing apoptosis, the created CQD-Ce6-HA compound completely inhibited melanoma

skin cancer in a mouse model [95]. Some recently investigated CQD-based nanosystems are given in Table 5.

**Table 5.** Recent developments in CQD-based drug delivery systems.

| Nanosystem | Drug | Application | Reference |
|---|---|---|---|
| FA-Gd-CQD | Doxorubicin | Anti-cancer effect against Hela cell lines | [96] |
| QA-N-CQDs | Gemcitabine | Theranostic system for breast cancer cell lines | [97] |
| m-CQD-HA | Hyaluronic acid | Site-specific delivery of HA to chondrocytes | [98] |
| N-CQD-Insulin | Insulin | Oral delivery of insulin | [99] |
| Mn/CQD/SiO2-naproxen | Naproxen | Drug delivery and tracer system | [100] |

*4.4. Carbon Nanotubes*

Carbon nanotubes (CNTs) are one-dimensional structures with $sp^2$ hybridized carbon atoms that are hollow and cylindrical [101]. In 1991, Japanese scientist Sumio Iijima became the first person to discover CNTs. These tubes vary in diameter, length, chirality, and layer count [96]. CNTs are similar to graphene sheets, which are hexagonal networks of carbon atoms rolled up in specific directions based on their chirality. SWCNTs are produced by rolling up a single graphene layer, whereas multi-walled CTNs (MWCNT) are produced by rolling up multiple graphene sheets [16]. These tubes have a variety of appealing qualities because of their varying elasticities, strengths, and rigidities [60]. The Young's modulus of elasticity determines how strong they are (ratio of stress to strain). CNTs are thought to have a tensile strength of 100 GPa. These nanotubes lose their flexibility permanently as a result of a process known as plastic deformation that takes place under extreme stress. It was discovered that the specific Young's modulus values of MWCNTs are lower than Young's modulus values of SWCNTs [102]. CNTs have a generally high length-to-diameter aspect ratio; for SWCNTs, the diameters typically range from 0.4 to 2.5 nm, while the lengths can range from 20 to 1000 nm, whereas for MWCNTs, the diameters range from 1.4 to 100 nm (thousands of times thinner than a human hair) and the lengths range from 1 to 500 nm [103]. CNTs have a specific internal volume because of their tubular form, which can be exploited to house particular functional moieties. Additionally, CNTs are more reactive than pristine graphene due to the existence of the curvature in their structure; thus, chemical functionalization of their surface becomes easier [104].

*4.5. A. Single-Walled Carbon Nanotubes*

SWCNTs are essentially graphene sheets that have been folded and extended. Patterns on a graphene sheet can be affected by both the sheet's diameter and its C-C orientation. As a result of their powerful interactions, single-wall carbon nanotubes have limited solubility and are difficult to disperse in the aqueous phase. It has been suggested that the SWCNTs can undergo chemical alteration in order to make them more soluble in water and better disperse themselves throughout it in order to boost their potential applications in biomedicine and pharmaceuticals [101,105].

*4.6. B. Multiple-Walled Carbon Nanotubes*

The electric and chemical properties of multi-walled carbon nanotubes (MWCNTs) are more complex due to the presence of many layers of graphene sheets that compose their structure. Nanotubes of this sort can range in size anywhere from 5 to 50 nm [106]. The structural complexity and variety in this class are a result of wrapping one CNT around another, with an interlayer dispersion of 3.4 Å. MWCNTs are, therefore, not as well understood. If the structure is only slightly altered, the desirable material qualities might be lost [107].

### 4.6.1. Synthesis

As the initial CNT synthesis process, the arc discharge method is well-known and frequently utilized [108]. In the arc discharge mechanism, two electrodes made of exceedingly pure graphite are spaced 2–3 mm apart. In addition, one of the electrodes contains traces of metal catalysts such as Fe, Co, Ni, or Mo. In a He environment, the voltage between the reaction chamber electrodes is then used to generate a DC arc discharge. The graphite and metal catalyst evaporate and condense as the discharge is triggered, resulting in carbon soot containing both SWCNTs and MWCNTs collecting on the cathode and/or reactor walls [104,109].

Laser ablation is another process used to create nanotubes. A laser procedure is used to create nanotubes, as the name suggests. The more sophisticated laser ablation technique is used to overcome the drawbacks of the electric arc technique. Both SWCNTs and MWCNTs can be produced using this method. The regular use of this method results in the synthesis of more nanotubes. A pulsed laser is used to ablate the carbon target during the synthetic process. In the presence of an inert gas and a catalyst, pulsed laser ablation of the graphite target is carried out in this manner [110].

The process of chemical vapor deposition is another method that is frequently applied to generate CNTs. Alcohols, methane, ethane, acetylene, and other hydrocarbon precursors are broken down in the presence of a metallic catalyst at high temperatures. When a reaction gas is pushed through a flow furnace at a temperature of 1000 °C, hydrocarbon molecules disintegrate into active carbon species on the catalyst surface and infiltrate the metal catalyst. As the broken-down hydrocarbons permeate the metal, the CNT develops between the substrate and the catalyst [31,110].

### 4.6.2. Functionalization

Pristine CNTs cannot be used for drug administration because they are insoluble in biological fluids when they are not functionalized with organic molecules. Intact CNTs form tightly packed bundles as a result of van der Waals interactions, rendering them insoluble in common solvents such as water [110]. Examples of techniques for improving the solubility of CNTs include copolymerization, organic functionalization, and acid treatment to provide carboxylic end groups. During the process of direct covalent sidewall functionalization, the formation of a covalent bond causes a change in the carbon hybridization from $sp^2$ to $sp^3$, as well as a concomitant loss of the p-conjugation that is characteristic of the aromatic rings of graphene. This loss of p-conjugation is due to the fact that the $sp^3$ carbon hybridization is more stable than the $sp^2$ carbon hybridization [111]. This mechanism can be produced by the atoms at the ends of the CNTs as well as in the defects of the CNTs by reacting with certain substances that have a high chemical reactivity. As a result of the polar or non-polar groups that are grafted onto the surface, the functionalization process allows carbon nanotubes (CNTs) to disperse in solvents, which is one of the essential advantages of the process [112]. PEGylated phospholipids are an example of the kind of noncovalent coating that can be applied to SWCNTs [113]. Using linear/branched PEGs of different molecular weights, Liu et al. (2008) [114] produced noncovalently functionalized SWCNTs to test the long-term effects of CNTs given intravenously to mice. Although, the weak bonding provided by hydrophobic contacts, van der Waals interactions, and other interactions limits the capacity of non-covalent functionalization to transport drugs, which can improve the dispersibility of CNTs [114].

### 4.6.3. Applications

Since nanotubes are hydrophobic, their even dispersion is difficult because of their limited solubility in aqueous solutions. It becomes difficult to uniformly integrate them with the API (active pharmaceutical ingredient). Various methods for dispersing CNTs in water and non-aqueous liquids have been recently investigated, with a focus on both physical difficulties and chemical strategies [16]. Examples of physical strategies include radiation, plasma treatment, and ultrasound. Additionally, a variety of materials have been

used to distribute CNTs, such as inorganic monovalent salts, inorganic peroxides, mineral acids, salts of organic acids, aromatic compounds, biomolecules, and polymers. As a result, in order for them to perform the function of carriers, they must go through the previously mentioned alterations [34].

Theranostics

CNTs have opened the door to research into their potential as theranostic and drug delivery system carriers due to their easy cell membrane permeability [114]. In 2020, Golubewa et al. [115] reported that glioblastoma cells had been shown to be effective in accumulating SWCNTs. The resulting large agglomerates enabled the photo-induced destruction of cancer cells by picosecond laser irradiation. Additionally, they used the CARS (Coherent anti-Stokes Raman scattering) imaging method and showed that it is viable for both the visualization of cancer cells and its photo-induced SWCNT-conditioned destruction, providing compelling evidence that this technology is helpful for nano theranostics [115]. In a different study, Song et al. (2016) [116] developed a carbon nanotube ring with gold nanoparticle coatings (CNTR@AuNPs). This CNTR bundle nanostructure was developed as a photoacoustic (PA) contrast agent for image-guided cancer therapy, as well as a cancer cell detection probe. It was embedded in a gap of closely spaced AuNPs. The CNTR@AuNP displayed noticeably greater optical and Raman signals when compared to a CNTR coated with an entire Au shell (CNTR@AuNS) and pure CNT@AuNP. The significantly increased PA signal and photothermal conversion capacity of CNTR@AuNP were effectively used for imaging and image-guided cancer therapy in two tumor xenograft models [116].

Biosensing

The detection of biomolecules with high precision is crucial in the diagnosis of various diseases for making timely decisions regarding pharmacotherapy. CNTs have a promising role in designing such technologies [117].

Glucose oxidase (GOx)-based catalysts were developed by Christwardana et al. in 2017 [118] and are capable of measuring glucose in the widest concentration range, accurately calculating the blood glucose concentration and enhancing the sensitivity of the glucose biosensor. A hybrid composite material with CNT reinforcement was generated using this catalyst. According to the analyses, a hybrid material modified with CNTs that has good chemical stability and high catalytic activity can be used to make a high-performance glucose biosensor with leading attributes. An excellent glucose sensitivity of 47.83 $\mu Acm^{-2}$ $mM^{-1}$, a low Michaelis–Menten constant of 2.2 mM, and broad glucose concentration detection were all displayed by the catalyst [118]. Another study designed reusable, inexpensive, and efficient lung cancer immuno-sensing biosensors using graphene oxide (GO) and carbon nanotubes (CNTs). Using CNT-based nanocomposites, the presence of the biomarker (hTERT) was found in the concentration range of 5 ng mL$^{-1}$ to 50 ng mL$^{-1}$. Due to the favorable properties (peak current, stability, repeatability, and reproducibility) acquired by employing the CNT, the simultaneous bianalyte detection of biomarkers (MAGE A2/A11) expressed in serum was also demonstrated [119].

Drug Delivery

CNTs have been extensively investigated for target-specific drug delivery due to their ease of fabrication and biocompatible properties. Drug loading into the end-capped CNTs can be accomplished both during and after CNT synthesis [120]. The diameter, surface tension, reactiveness, melting temperature, and sensitivity of the CNTs are all essential factors in drug loading. After synthesis, drug loading can be achieved by electrically opening the ends of the CNTs, corroding the angled end parts of the tube with an acid, or oxidizing the CNT with $CO_2$ [121].

CNTs are potential carriers in brain-targeted drug delivery. MWCNTCOOHs were utilized to carry oxaliplatin to the brain for the treatment of orthotopic gliomas. The

TAT-PEI-B copolymer was created by combining the BBB piercing peptide transcriptional activator, the glioma-targeting chemical biotin, and polyethyleneimine. Afterwards, the TAT-PEI-B-MWCNT-COOH@OXA complex was formed by mixing MWCNT-COOH and oxaliplatin. Characterization studies revealed that MWCNT had OXA loaded on its surface. The in vitro investigations included rat glioma cells as well as human glioma cells (U87 and U251). Mice with an artificially produced orthopedic brain glioma were used in in vivo experiments. This nanosystem was found to be more cytotoxic than free oxaliplatin [122]. The activity of functionalized CNTs has also been tested against pathogenic bacteria. The antibacterial activity of MWCNTs conjugated to ionic and non-ionic surfactants against *E. coli* growth was studied. Gram-negative *E. coli* was found to be more toxic to MWC-NTs coated with cationic surfactants. Anionic surfactant-coated MWCNTs were more hazardous than non-ionic surfactant-coated MWCNTs due to the combined toxicity of the surfactant [123]. Some promising studies on NCTs are given in Table 6.

**Table 6.** Recent developments in CNT-based drug delivery.

| Nanosystem | Drug | Application | Reference |
|---|---|---|---|
| MWCNTs/Gemcitabine (Ge)/Lentinan-Le | Gemcitabine-Lentinan | Chemophotothermal therapy for cancer | [124] |
| PEGylated-MWCNTs | Dexamethasone | CNS delivery of dexamethasone for ischemic stroke | [125] |
| Magnetic MWCNTs-Epirubicin | Epirubicin | Site-specific and prolonged release of epirubicin for bladder cancer | [126] |
| CA/SWCNT-Gl | Curcumin | Antibacterial activity against *B. cereus* and *E. coli* | [127] |
| SWCNT-COOH- Levodopa | Levodopa | pH-dependent CNS release of levodopa | [128] |
| SWCNT-Acetylcholine | Acetylcholine | CNS delivery of Ach, restoring memory in mice | [129] |

*4.7. Nanodiamonds*

Nanodiamonds (ND) are an allotrope of carbon (sp$^3$ hybrid) with diameters of 2–10 nm and possessing a high surface area. They are diamond-shaped nanocrystals with tetrahedral carbon atoms organized in a cubic 3D lattice, and they have the electrical properties of diamond (high charge carrier mobility and breakdown field). NDs have a graphite shell that resembles an onion. The three types of NDs include diamondoids (1–2 nm), ultrananocrystalline particles (2–10 nm), and nanocrystalline particles (1–150 nm). The nitrogen-vacant positions at the center of nanodiamonds make them fluorescent; thus, NDs are suited explicitly for bioimaging purposes [60]. The large surface area, high thermal conductivity, inherent biocompatibility, and ease of functionalization make NDs suitable candidates for biomedical applications. NDs, similar to any other carbon nanostructures, are hydrophobic and thus possess a tendency to agglomerate in aqueous solutions with poor dispersibility [102]. Various techniques such as ultrasonic dispersion and mechanical grinding have been employed to form stable dispersions. However, surface functionalization is the most effective way of modifying their physicochemical and biological properties. Hydroxyl, carboxyl, sulfur, amino, and anhydride groups can all be used to modify the surface of NDs [14,102]. The functionalized NDs can be covalently linked to various other polymers and biological macromolecules to prepare NDs of particular interest [130].

4.7.1. Synthesis

The classification of diamonds is still applicable to nanodiamonds, which are created by fragmenting enormous bulk diamonds using top-down techniques. High pressure and high temperature (HPHT), dynamic synthesis employing detonation techniques, ball milling of HPHT diamond microcrystals, chemical vapor deposition, and laser ablation are the typical synthetic processes for NDs [131]. At room temperature, microcrystalline diamonds are compressed between 0.2 and 0.8 GPa pressure. The breaking of the crystals occurs in three stages as the pressure is increased: (1) fracturing of corners and edges, (2) cracking of

the lattice planes, and (3) fine-tuning of particle disorder [132]. Chemical vapor deposition and high pressure and high temperature (HPHT) methods are cost-effective and can be used for large-scale synthesis of NDs [133].

In the detonation process, graphite or any carbon precursor is packed inside a chamber with explosive material. The shock waves induced by the explosion increase the pressure inside the chamber by up to 100 Gpa and the temperature to more than 1700 °C. These extreme conditions enhance the rate of conversion of graphite into nanodiamonds [134]. The HPHT process can result in synthetic diamonds with up to 300 ppm of nitrogen. A diamond carbon atom is forced out of the structure when these diamond particles are exposed to high-energy radiation, causing damage and vacancies (V). These vacancies travel toward the nitrogen (N) centers during thermal annealing, creating nitrogen-vacancy (N-V) color defect centers. The emission of fluorescence is caused by these centers. Consequently, a wealth of research has been undertaken on high-pressure, high-temperature synthetic diamonds for their applications in bioimaging [102,130]. The dissociation of ethanol vapor is promoted in the chemical vapor deposition technique in a microplasma running at atmospheric pressure and neutral gas temperatures of 100 °C. The ND surfaces can change, and the non-diamond phases can be etched when H is added to the plasma atmosphere. The fabricated NDs range in size from 2 to 5 nm and contain cubic and lonsdaleite crystal structures. NDs have a core and shell structure at their core, but their size, shape, and surface properties depend on the circumstances of the explosion and the conditions during purification [135]. A portion of the surface shell is made up of graphitic structures, and the inert core is made up of diamond carbon. A wide variety of other functional groups, such as ether, ketone, anhydride, lactone, carboxyl, and many others, may also be present on the surface of these ND particles. By using oxidative or reductive reactions to homogenize the surface chemistry of NDs, uniform surface properties can be obtained [136].

### 4.7.2. Functionalization

It is essential to keep in mind that there might be significant variation across the various reactants or even within batches. The situation is further complicated by the fact that it is challenging to characterize and quantify organic functions, often yielding conflicting outcomes [137]. NDs are often subjected to a process of surface homogenization first before any surface functionalization because of these reasons. Such modified NDs can be employed for tissue scaffolding or drug administration by physically adsorbing or chemically grafting tiny molecular compounds or nucleic acids onto them [138]. To produce silanized NDs, NDs are first oxidized and carboxylated, and then surface modification is employed (NDs-NH2). The ND surface can be oxidized when the amorphous/graphitic coating of the diamond core is removed using strongly oxidizing substances such as potent acids, singlet oxygen in NaOH, potent ozone, and air treated with a catalyst [139]. DNA sequences and polymeric molecules are attached to the ND surface by electrostatic contact. The most popular and efficient method of functionalization is the esterification of the hydroxyl groups through high-temperature -OH carboxylation, which is accomplished by succinic anhydride. Another approach for derivatizing NDs involves covalently anchoring dyes and receptors to siloxanes, which have been successfully employed for drug delivery [140].

Surface functionalization has also become a cutting-edge method of ND aggregate size reduction. By functionalizing with long alkyl chains, the particle size can be reduced from 15 μm to 150–450 nm. Compared to their native form, the long alkyl chain-modified NDs are highly dispersible in organic solvents [131,140].

### 4.7.3. Applications
Bioimaging

The unique optical, chemical, and biocompatible properties of NDs make them the preferred candidates for developing bioimaging technologies. As discussed above, the nitrogen vacancies in NDs make them fluorescent and are used to detect specific biomolecules [140]. In comparison to commonly employed fluorophores, NDs have a higher advantage for

long-term cellular imaging due to the extraordinary photostability of (N-V)- defect centers. In contrast to similarly sized red fluorescent polystyrene nanospheres, which photobleach during the first 30 min of photoexcitation, the red fluorescent NDs (100 nm) exhibit no photobleaching after being constantly exposed to a 100 W lamp for 480 min. ND fluorescence can withstand photoblinking for up to 1 millisecond [141]. In 2010, Mohan et al. studied long-term in vivo imaging in *C. elegans* using fluorescent NDs. The colloidal solution of NDs was fed to *C. elegans,* while microinjection of NDs was administered into the gonads of *C. elegans.* The cellular development process of *C. elegans* was studied for several days, and bioimages of the GI tract and tissues with high brightness were obtained. No cytotoxicity or genotoxicity was reported in the long-term study [142]. In another study, single high-dose fluorescent NDs were administered to rats. Bio-distribution and pharmacokinetics of NDs were studied. The liver and spleen were significant sites of distribution, and high-contrast fluorescent and SEM images were obtained with no cytotoxicity [143]. In 2013, Merson et al. [144] prepared SiV-containing NDs using the CVD process and silicon ion implantation for optical imaging of stem cells. Zurbuchen et al. (2013) [145] developed fluorescent NDs with N vacancies. ND cellular uptake was validated by optical microscopy, which revealed significant in vitro fluorescence.

Gene Therapy

NDs are ideal carriers of genetic material to bodily tissues due to their high surface area and tunable surface chemistry. In 2012, Alhaddad et al. [102] synthesized siRNA (short interfering RNA)-loaded NDs coated with polyallylamine (PAH) and polyethyleneimine (PEI). This formulation was administered to Ewing sarcoma cells. The PEI-coated and siRNA-loaded NDs increased the penetration of the target gene into the Ewing sarcoma cells, and the silencing of oncogene EWS-Fli1 was observed [146]. In another study, a siRNA and ND complex was obtained through electrostatic conjugation. The penetration and delivery of siRNA inside hepatocellular carcinoma cells were reported to be higher than the activity of liposomal formulations [147]. Another interesting study was conducted by Al Qtaish et al. in 2022 [148], where hybrids of NDs and niosomes were prepared, called nanodiasomes. The transaction efficiency of nanodiasomes was studied on HEK-293 cells, and increased clathrin-mediated endocytosis of gene-loaded nanodiasomes was observed [148]. NDs can also find applications in peptide and small molecule delivery using techniques such as enzyme immobilization and CRISPR-Cas9 [149].

Drug Delivery

Apart from the high loading capacity of NDs, the release pattern of drugs from NDs can also be tuned as desired. NDs have been functionalized with polymers and lipophilic moieties to improve drug delivery to deep-seated tumors and the CNS. To improve their aqueous dispersibility and cell transport, chemotherapy drugs such as 4-hydroxytamoxifen and purvalanol-A can be adsorbed onto the surface of NDs [150].

In 2022, Tian et al. [151] prepared diazoxide-containing florescent NDs. This system was aimed at simultaneous imaging as well as drug delivery to HeLa cells. Drug-loaded fluorescent NDs enhanced the delivery of drug inside the cells, sustained their release by up to 72 h, and also measured the potential of the drug to generate ROS inside HeLa cells [151]. In an anticancer study, chitosan-coated NDs were prepared to provide steric stability and mucoadhesiveness to NDs. The chitosan-coated NDs were loaded with doxorubicin and further coated with pentasodium tripolyphosphate. A potent cytotoxic effect was observed on bladder cancer cell lines HT-1197 [152]. In 2022, Zhong et al. [153] fabricated iron-nanoparticles encapsulated within graphene shells that were loaded with ferulic acid. The prepared nanoformulation was injected subcutaneously into diabetic BALB/c mice. Ferulic acid-loaded nanoparticles were found to be effective in lowering the blood glucose level in mice and sustained the drug release (8.75 mg/g/day) over 30 days [153]. Some recent studies on NDs are given in Table 7.

**Table 7.** Recent advances in nanodiamond-based drug delivery.

| Nanosystem | Drug | Application | Reference |
|---|---|---|---|
| ND-DOX | Doxorubicin | Inducing cytotoxicity in colorectal carcinoma cells | [130] |
| ND-Alen | Alendronate | Enhancing osteogenic effect in bones | [154] |
| ND-4-AP | 4-Aminopyridine | CNS delivery of epileptogenic drugs | [155] |
| ND-Aml | Amlodipine | CNS delivery of amlodipine for neurological disorders | [156] |
| ND-PEI-$Fe_3O_4$Amox | Amoxycillin | Magnet-responsive delivery of antibiotics at the target site | [157] |

## 5. Mechanism of Cellular Uptake of Carbon Nanostructures

Though the uptake of carbon nanostructures largely depends on the surface chemistry, size, and functionalization, it has been found that nanostructures with size < 50 nm undergo internalization by diffusion and energy-independent uptake. Nanostructures with a size range of 50–100 nm undergo endocytosis. This phenomenon has extensively been studied and proved in carbon nanotubes [158].

Carbon nanostructures have so far been administered to animal models by abdominal, subcutaneous, intravenous, and intra-peritoneal routes. Smaller particles are readily distributed to body tissues. The fate of circulating nanostructures depends on the surface chemistry of the particles [159]. It has been reported that PEGylation of carbon nanostructures retards the uptake of carbon nanostructures by the reticuloendothelial system, increasing their residence time in the body. The primary cellular uptake route for carbon nanostructures is receptor-mediated endocytosis, also referred to as clathrin-mediated endocytosis [160]. Clathrin, a triskelion membrane protein, is responsible for forming capsule-like invaginations upon encountering particles of a specific size range. The ligands on the surface of nanostructures bind to the active sites of the clathrin receptor, causing the membrane to change shape (Figure 3). A vacuole is formed, causing the nanostructures to enter inside the cell by packing them into small vesicles called early endosomes. The early endosome then matures into late endosomes and eventually fuses with a lysosome, releasing the internalized content into the cytoplasm [158]. In 2017, Li et al. [161] demonstrated clathrin-mediated endocytosis of functionalized SWCNTs. In another study, clathrin-mediated endocytosis was used to internalize recombinant Ricin-A chain functionalized multi-walled carbon nanotubes in human cervical cancer HeLa cells [162].

Caveolae-mediated endocytosis is a type of receptor-mediated endocytosis that is clathrin-independent. Caveolae are flask-shaped membrane invaginations or pits that are 50–80 nm in size and are responsible for pinching out small pieces of the cell membrane after packing external molecules/particles. These molecules/particles are then packed inside small vacuoles and transported toward the cytoplasm. The vesicle so-formed is called a caveosome, which then fuses with an endosome, matures to a late endosome, and undergoes lysosomal fusion and degradation. This process is comparatively slower than clathrin-mediated endocytosis [163]. In 2018, Chen et al. [164] prepared COOH-functionalized carbon nanoparticles tagged with fluorescent dye. Caveolae-mediated endocytosis was observed in A549 mammalian cell lines. These two phenomena can easily be observed through epifluorescence and confocal microscopy. The particles with sizes > 150 nm are not well internalized by these two processes; therefore, they undergo internalization through clathrin and caveolae-independent pathways. Carbon nanostructures with a size < 50 nm can penetrate the cell membrane by simple diffusion, but this also depends on the surface characteristics of the nanostructures [165]. The process of cell internalization by simple diffusion occurs through three steps: (A) floating of nanostructures on the cell membrane, (B) orientation of lipophilic heads of nanostructures towards cell membrane, and (C) sliding and penetration of nanostructures. This process is called needle-like penetration in the case of carbon nanotubes. The cell internalization processes such as

macropinocytosis, phagocytosis, and other active processes are less reported and are restricted to distinctively designed carbon nanostructures [166].

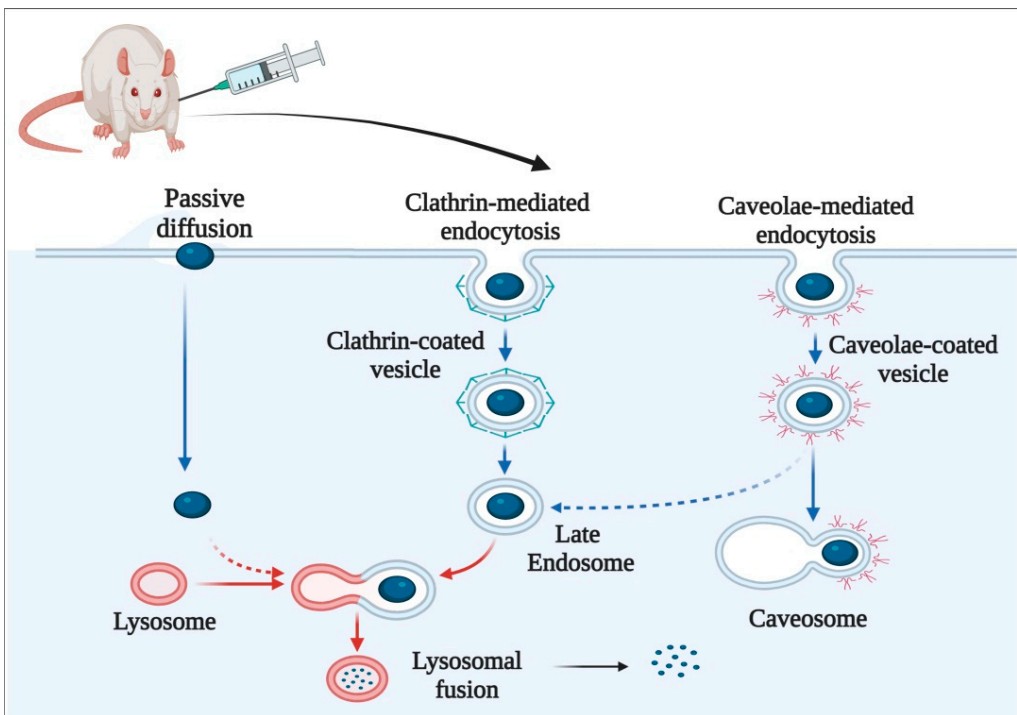

**Figure 3.** Major cellular pathways for internalization of carbon nanostructures.

## 6. Pharmacokinetic Considerations

The pharmacokinetic properties of carbon-based nanostructures largely depend on their size, shape, molecular arrangement, functionalization, solubility, and lipophilicity. Thus, their distribution in various body tissues, cellular uptake, accumulation, recognition by the reticuloendothelial system, degradation, and excretion are major fields of study to design systems with desired pharmacokinetic properties [167].

The carbon nanostructures used for biomedical purposes are either functionalized or dispersed with suitable surfactants. Such optimized systems, when administered in vivo, have been reported to undergo excretion depending on their physicochemical characteristics. Otherwise, new carbon nanostructures, as a result of their hydrophobicity, have the tendency to aggregate and accumulate in body tissues. The metabolism of nanostructures also depends on the microenvironment they are provided with. The metabolism of CNTs by neutrophil peroxidase in the pulmonary tissues has been reported [168]. Carbon nanodots have also been shown to disintegrate via peroxide catalysis in the presence of lipase [169]. Fluorescence spectroscopy and SDS-PAGE electrophoresis detected the binding of blood proteins albumin, transferrin, Ig, and fibrinogen to the surface of carbon nanostructures. This phenomenon restricted the cellular uptake of nanostructures, as did the tissue distribution [170]. Carbon nanostructures undergo extensive defunctionalization reactions in the hepatocytes, leading to the conversion of functionalized nanostructures into pristine nanostructures with aggregation and accumulation potential. This transformation has been reported less in covalently functionalized nanostructures than non-covalently functionalized nanostructures. Nitrogen-doped nanostructures undergo the least amount of defunctionalization reactions in the liver [102]. Their carbon skeleton has a high stability, due to which the integrity of the skeleton is maintained. This could be the significant reason for target organ toxicity, where carbon nanostructures accumulate in specific organ for a long duration, release ROS, and initiate apoptosis [171].

Radioisotope labeling, Raman spectroscopy, intrinsic NIR fluorescence, and microscopic observation are among the major techniques to study in vivo biodistribution of

carbon nanostructures. Among these techniques, radioisotope labeling is a quantitative method. Radioisotopes are attached to the surface of carbon nanostructures and administered through various routes. Commonly used radioisotopes for this purpose are $^{14}C$, $^{125}I$, $^{86}Y$, $^{99}mTc$, and $^{111}In$. Studies are conducted for a pre-determined period, and the organs of animals are collected for analysis. Liquid scintillation counters and $\gamma$-counters are used for the tracing and quantitative analysis of the distribution of carbon nanostructures. For convenience, carbon nanostructures can also be visualized in tissues through TEM (transmission electron microscopy), as these nanostructures have distinguishable structures from cell organelles [172].

## 7. Toxicity of Carbon Nanostructures

Carbon-based nanomaterials cannot be considered as completely inert. These nanostructures have the potential to interfere with cell-signaling pathways leading to the production of reactive oxygen species, leading to toxicity and cell death. Novel methods of synthesis and functionalization should be investigated to fabricate safer drug-delivery cargoes [173].

Carbon-based nanoformulations have been investigated thoroughly in vitro and in animal models, with positive results. There are still concerns about their toxicity potential in humans, as not much safety data are available. Cancer-based nanostructures have a high tendency to aggregate and can accumulate in body tissues. These structures penetrate the cell membrane by various uptake pathways [170,173]. Long-term animal studies are needed to assess the possibility of accumulation in tissues, which could lead to target organ harm. This effect, when intentionally developed against deep-seated tumors, can be explored for anti-cancer therapy, but still, target specificity is a significant challenge. There is evidence that carbon-based nanostructures behave similar to asbestos, causing toxicity in the lungs and fibrosis. The production of reactive oxygen species (ROS), lysosomal damage, and mitochondrial dysfunction all contribute to cytotoxicity, which eventually lead to necrosis or apoptosis [174]. According to the National Institute of Occupational Safety and Health, all carbon-based nanomaterials should be considered occupational respiratory risks, and the recommended exposure control limit is 1 g/m3 elemental carbon over an 8-h exposure period [175].

The fundamental parameters of these structures, such as shape, size, functionalization, and aggregation index, need to be studied for designing systems with real-life applications in clinical settings [176]. In 2014, Khalid et al. [177] investigated the toxicity of functionalized multiwalled carbon nanotubes on Saos cell lines and found no toxicity up to a concentration of 1000g/mL. According to Singh et al. (2006) [178], intravenous injection of water-soluble ammonium-functionalized single-walled carbon nanotubes in mice caused no harm; CNTs were least deposited in the spleen and liver and were quickly eliminated in urine. In another study by Wang et al. in 2004 [179], hydroxyl functionalized single-walled CNTs were administered intraperitoneally in mice and it was found that CNTs were excreted in urine within 18 days, while the least accumulation of CNTs was reported in the spleen and liver. Although lipid-functionalized and co-polymerized nanostructures show less toxicity, comparative toxicity studies of functionalized vs. non-functionalized nanomaterials need to be conducted to assess the relative added benefit provided by functionalization [180].

## 8. Recent Patents

Many formulation scientists have developed technologies for delivery through novel approaches. Some of the recently filed/granted international patents based on innovations in carbon-based nanomaterials are given in Table 8.

**Table 8.** List of patents on carbon nanostructures.

| Patent No. | Publication Date | Classification | Title | Inventors | Organization | Reference |
|---|---|---|---|---|---|---|
| EP1558524B1 | 16 January 2019 | Carbon nanotubes | CNTs synthesis by CVD | Milo Sebastian Peter Shaffer, Alan H. Windle, et al. | Zeon corp. | [104] |
| IN202041056414 | 1 July 2022 | Carbon nanodots | Ocular drug delivery and tracking comprising photo-luminescent CNTs | K. Sudhakaraprasad, Kumara. B. N., et al. | Yenepoya (Deemed To Be University) | [181] |
| WO 2017/029501 Al | 23 February 2017 | Carbon nanodots | Conjugated quantum dots | Naasani, Imad | NANOCO TECHNOLOGIES LTD | [182] |
| WO/2022/160055 | 4 August 2022 | Carbon nanohorns | Method of manufacturing carbon nanohorns and the carbon nanohorns thus produced | Braidy, Nadi Casteignau, Fanny, et al. | UNIVERSITÉ DE MONTRÉAL | [183] |
| US20200385272 | 6 July 2020 | Carbon nano-onion | Porosity for catenated carbon nano-onions and surface area | Danny Cross, Larry Herbert Kirby, Thomas Frank Bailey, et al. | Structured Nano Carbon LLC | [184] |
| AU2021104363 | 19 August 2021 | Carbon nanotubes | CNTs for camptothecin targeted release in cancer chemotherapy | Bhowmick, Mithun, Bhowmick, Pratibha, et al. | Individual | [185] |
| US20200276126 | 3 August 2020 | Carbon nanodiamonds | Polydopamine-encapsulated nanodiamonds and methods | Haksung Jung and Keir Cajal Neuman | NIH, USA | [186] |

## 9. Challenges and Future Perspectives

Biomedical research on carbon nanostructures has been expanding since the discovery of fullerene. As a versatile member of materials science, carbon and its derivatives find application in diagnostics, drug delivery, gene therapy, immunobiotechnology, and biosensing. Still, many challenges need to be addressed to lead to some breakthroughs. The commercial production of carbon nanostructures is the first challenge. The studies that have been explored so far are either on a lab scale or a pilot scale. For the wider applicability and study of carbon nanostructures, optimized methods of synthesis need to be developed. These methods should be cost-effective, eco-friendly, and high yielding. The generation of toxic residues with the production of carbon-based nanomaterials is a big issue in ecotoxicity. In vitro and in vivo studies have confirmed that inhalation of these materials are responsible for causing lung fibrosis. Thus, developing regulatory protocols for the synthesis and commercialization of carbon nanoproducts is a significant challenge for regulatory bodies such as the USFDA.

Another hurdle in the development of carbon nanostructure-based drug products is poor aqueous solubility/dispersibility. Better synthesis, characterization, functionalization, and refinement techniques need to be developed for controlling the particle size and surface area of carbon nanostructures to significantly affect the formation of stable dispersions. Novel functionalization methods can be explored to produce nanostructures with desired biopharmaceutical characteristics. Apart from this, the conjugation or hybridization of nanostructures with other polymers such as pH-sensitive polymers, magnet-sensitive polymers, and redox-sensitive polymers can be studied if the resultant systems possess better responsiveness in biological media. Thus, specialized nanocarriers can be designed for the targeted delivery of drugs to the brain, liver, lungs, bones, and tumors. The drug-delivery systems based on carbon nanostructures are solely studied in in vitro models and in vivo studies on rodents. Data are available on the toxicity of carbon nanostructures. This phenomenon has also been explored for cytotoxic effects in tumor cells, but with

poor selectivity. Long-term toxicity studies should be performed, employing each type of carbon nanostructure as per OECD guidelines. The molecular mechanisms involving cytotoxicity should be studied so as to develop suitable functionalization routes of carbon nanostructures with enhanced in vivo safety.

Thus, it can be concluded that functionalization is a crucial factor in deciding the fate of carbon-based nanostructures. With advancements in green chemistry, nanostructures with high biocompatibility can be synthesized for applications in tissue engineering, biosensing, bioimaging, gene therapy, theranostics, and drug delivery. The data in the public domain should be analyzed for effective decision-making. Once carbon nanoformulations are safely tested and documented in animals, the studies should be translated to primates and then to humans for critically evaluating the therapeutic potential of carbon nanostructures in diseases such as cancer and neurodegenerative and metabolic disorders.

## 10. Conclusions

The adaptability of carbon nanostructures as platforms with the possibilities for diverse targeting, therapeutics, and diagnostics is highlighted in this paper. These features can be combined so that a single medication can detect, select, deliver cargo, and elicit explicit reactions. This is possible because carbon nanostructures have high functionalization potential. The ideal nanosystem must be observable, have specific targeting functional groups, and generate an explicit biological reaction. All of this is achievable with carbon-coated nanoparticles, making them interesting nanovectors for medication delivery and imaging contrast agents. Carbon nanostructures have the potential to be used in biomedical applications. Continued investment and research in this subject are required to achieve an accurate insight of their benefit vs. any risk that they may bring. The research presented here suggests that carbon nanostructures have a promising future in healthcare applications.

**Author Contributions:** Conceptualization, J.S. and P.N., writing—original draft preparation, J.S., P.N., G.S. and M.K.; writing—review and editing, R.R.S., M.K.K., G.S. and M.K.; supervision, R.R.S., M.K.K., G.S. and M.K. All authors have read and agreed to the published version of the manuscript.

**Funding:** This research received no external funding.

**Institutional Review Board Statement:** Not applicable.

**Informed Consent Statement:** Not applicable.

**Data Availability Statement:** Not applicable.

**Acknowledgments:** The figures provided in this article were created with Biorender.com (accessed on 30 November 2022).

**Conflicts of Interest:** The authors declare no conflict of interest.

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
