# Peer review of "Carbon Nanostructures as Therapeutic Cargoes: Recent Developments and Challenges"

_carbon, 2022_

Round 1

Reviewer 1 Report

This is a well-organized review paper. This paper summarizes all the previous reports of carbon nanostructures as therapeutic cargoes and is potentially useful to readers in biomedical field. After reading this manuscript from the beginning to the end, I cannot find serious weakness for this study. I have just one minor suggestion for this review paper: there is a review article by Professor Truskewycz entitled “Carbon Dot Therapeutic Platforms: Administration, Distribution, Metabolism, Excretion, Toxicity, and Therapeutic Potential" that should have been discussed and referenced in the manuscript (Small 2022, 18, 2106342). Similarly, there is two books by Grumezescu on "Nanomaterials for Drug Delivery and Therapy" and Dave, Das and Ghosh on “Advanced Nanomaterials for Point of Care Diagnosis and Therapy” that should have been referenced (ISBN: 978-0-12-816505-8 and 978-0-323-85725-3), as well as two recent articles by the Cai and Yeung groups in Adv. Healthcare Mater. 2022, 11, 21022, 134806 and Chem. Eng. J. 2022, 435, 134466. Given the above suggestions, I would recommend this paper for acceptance after minor revision.

Author Response

Response of Reviewer 1.

Thank you for your review of our paper. We have responded to each of your points below.

  1. Comments and Suggestions for Authors

There is a review article by Professor Truskewycz entitled “Carbon Dot Therapeutic Platforms: Administration, Distribution, Metabolism, Excretion, Toxicity, and Therapeutic Potential" that should have been discussed and referenced in the manuscript (Small 2022, 18, 2106342). Similarly, there is two books by Grumezescu on "Nanomaterials for Drug Delivery and Therapy" and Dave, Das and Ghosh on “Advanced Nanomaterials for Point of Care Diagnosis and Therapy” that should have been referenced (ISBN: 978-0-12-816505-8 and 978-0-323-85725-3), as well as two recent articles by the Cai and Yeung groups in Adv. Healthcare Mater. 2022, 11, 21022, 134806 and Chem. Eng. J. 2022, 435, 134466.

Response

Reference No. 77

Reference No.86

Reference No.153

Reference No.173

Reviewer 2 Report

Below are some of the specific comments that need to be addressed:

Line 6-10 and 11: In Affiliations, please use the same font size for affiliations and correspondence. Add space after Affiliations 1. Check the journal guidelines.

Line 32: Capitalize ‘s’ for Richard Smalley.  

Line 61: Please mention the abbreviation (NDs) after the nanodiamonds full form used in above line 60.

Line 67: Carbon nanotubes (CNTs), abbreviation already mentioned in line 59

Line 68: Abbreviation GR stands for what?

Line 106, 598: Please remove the space for ‘575〫C’.

Table 1-8: Please check the font size for all the tables through the manuscript. Check journal guidelines.

Line 206-208: Add references.

Line 210: Please enhance the readability of the sentence  ‘….in COVID-19 related hospitalization’. Also check for the spelling.

Line 209-216: Please reconstruct complete para using proper chemical terms.

Line 322: Remove ' from CQDs'.

Line 327-328: Add reference to ‘When exposed to light, these chemicals generate reactive oxygen species, which can kill bacteria and hence treat illnesses (ROS).’ Also, what ROS stands for?

Line 344: H2SO4 use subscript.

Line 418: Please change ‘….a hollow tube or cylindrical shape’ to ‘hollow cylindrical structures.’

Line 421-422: Please reconstruct the following sentence: ‘…have been rolled up in specific directions according to their chirality’.

Line 424; Add reference ‘These tubes have a variety of appealing qualities because of varying elasticity, strength, and rigidity.’

Line 439: Abbreviation SWCNTs already explained in section 4. Please be consistent.

Line 460: What ‘walls103.’ Stands for?

Line 491 and 493: Is SWNTs same as SWCNTs, if not also mention it in full form.

Line 500, 513 ‘API’, ‘CARS’; please mention full form since it’s the first-time abbreviations is used.

Line 508: Abbreviation already mention in line 489, please remove the repetitions.

Line 562: E. coli please write  in full form as well as in italics.

Line 570: Please mention what properties ‘….they have the electrical properties of diamond.’

Line572: Add space (1 to 2nm) as (1 to 2 nm).

Line 577: Remove space, . at the start of the sentence.

Line 581-582: Please add references for following sentence; ‘The hydroxyl, carboxyl, Sulphur, amino, and anhydride groups can all be used to modify the surface of NDs.’

Abbreviations: TEM

Line 786: NIOSH; stands for?

Please check extra spaces before the sentences, extra ,, throughout the manuscript.

References: Please check the journal style guideline to add the reference.

Authors should proofread the manuscript thoroughly to correct grammatical errors, long sentences and terms throughout the manuscript. Some part of the manuscript is well written whereas others are needs to be improved.

Author Response

Response of Reviewer 2.

Thank you for your review of our paper. We have responded to each of your points below.

Comments and Suggestions for Authors

  1. Line 6-10 and 11: In Affiliations, please use the same font size for affiliations and correspondence. Add space after Affiliations 1. Check the journal guidelines.
  2. Line 32: Capitalize ‘s’ for Richard Smalley.
  3. Line 61: Please mention the abbreviation (NDs) after the nanodiamonds full form used in above line 60.
  4. Line 67: Carbon nanotubes (CNTs), abbreviation already mentioned in line 59
  5. Line 68: Abbreviation GR stands for what?
  6. Line 106, 598: Please remove the space for ‘575〫C’.
  7. Table 1-8: Please check the font size for all the tables through the manuscript. Check journal guidelines.
  8. Line 206-208: Add references.
  9. Line 210: Please enhance the readability of the sentence ‘….in COVID-19 related hospitalization’. Also check for the spelling.
  10. Line 209-216: Please reconstruct complete para using proper chemical terms.
  11. Line 322: Remove ' from CQDs'.
  12. Line 327-328: Add reference to ‘When exposed to light, these chemicals generate reactive oxygen species, which can kill bacteria and hence treat illnesses (ROS).’ Also, what ROS stands for?
  13. Line 344: H2SO4 use subscript.
  14. Line 418: Please change ‘….a hollow tube or cylindrical shape’ to ‘hollow cylindrical structures.’
  15. Line 421-422: Please reconstruct the following sentence: ‘…have been rolled up in specific directions according to their chirality
  16. Line 424; Add reference ‘These tubes have a variety of appealing qualities because of varying elasticity, strength, and rigidity.’
  17. Line 439: Abbreviation SWCNTs already explained in section 4. Please be consistent.
  18. Line 460: What ‘walls103.’ Stands for?
  19. Line 491 and 493: Is SWNTs same as SWCNTs, if not also mention it in full form.
  20. Line 500, 513 ‘API’, ‘CARS’; please mention full form since it’s the first-time abbreviations is used.
  21. Line 508: Abbreviation already mention in line 489, please remove the repetitions.
  22. Line 562: E. coli please write in full form as well as in italics.
  23. Line 570: Please mention what properties ‘….they have the electrical properties of diamond.’
  24. Line572: Add space (1 to 2nm) as (1 to 2 nm).
  25. Line 577: Remove space, . at the start of the sentence.
  26. Line 581-582: Please add references for following sentence; ‘The hydroxyl, carboxyl, Sulphur, amino, and anhydride groups can all be used to modify the surface of NDs.’
  27. Abbreviations: TEM
  28. Line 786: NIOSH; stands for?
  29. Please check extra spaces before the sentences, extra ,, throughout the manuscript.
  30. References: Please check the journal style guideline to add the reference.

Responses

  1. Revised and highlighted
  2. Revised and highlighted
  3. Revised and highlighted
  4. Revised and highlighted
  5. It was Graphene (GP) Reactefied and highlighted
  6. Revised and highlighted
  7. Revised
  8. Revised and highlighted
  9. Revised and highlighted
  10. Revised
  11. Removed
  12. Reactive oxygen species (ROS), highlighted
  13. Revised and highlighted
  14. Revised and highlighted
  15. Revised and highlighted
  16. Added
  17. Revised
  18. Rectified
  19. SWNTs and SWCNTs both are same, Revised
  20. Mentioned
  21. Mentioned and highlighted
  22. Removed
  23. Mentioned
  24. Added
  25. Removed
  26. Revised and highlighted
  27. Mentioned abbriviation of TEM
  28. Mentioned
  29. Checked and Revised
  30. Revised as per guideline

Reviewer 3 Report

The manuscript entitled “Carbon Nanostructures as therapeutic cargoes: Recent developments and challenges” aims to review the physicochemical properties of carbon nanostructures, their categories, method of synthesis, various techniques of surface functionalization, major biomedical applications, mechanisms involving the cellular uptake of nanostructures, pharmacokinetic considerations, recent patents on carbon-based nanostructures in the biomedical field, major challenges and future perspectives. This topic is of high importance. The manuscript is overall well prepared. I have some minor suggestions.

The introduction should better explain the motivation for writing this paper.

Chapter 3, Green approaches… What about the use of waste for carbon material synthesis? It is an important green aspect of this topic.

Figure 2 seems incomplete. It should contain all materials elaborated in the manuscript, like nanodiamonds etc.

Line 693: mistake in section numbering. It should be 5, not 10.

Sections 6 and 7 should be given in more detail.

Please, check the reference style.

Author Response

Response of Reviewer 3.

Thank you for your review of our paper. We have responded to each of your points below.

Comments and Suggestions for Authors

  1. Chapter 3, Green approaches… What about the use of waste for carbon material synthesis? It is an important green aspect of this topic.
  2. Figure 2 seems incomplete. It should contain all materials elaborated in the manuscript, like nanodiamonds etc.
  3. Line 693: mistake in section numbering. It should be 5, not 10.
  4. Sections 6 and 7 should be given in more detail.
  5. Please, check the reference style.

Responses

  1. Revised and highlighted
  2. Rectified and mentioned Nanodiamond in the figure.2
  3. Rectified
  4. Added
  5. Revised as per guideline
